# Signatures of folded branches in the scanning gate microscopy of ballistic electronic cavities

Keith R. Fratus[1,2], Camille Le Calonnec[1,3], Rodolfo A. Jalabert[1], Guillaume Weick[1], Dietmar Weinmann[1*]

**1** Université de Strasbourg, CNRS, Institut de Physique et de Chimie des Matériaux de Strasbourg, UMR 7504, F-67000 Strasbourg, France
**2** HQS Quantum Simulations, Haid-und-Neu-Str. 7, 76131 Karlsruhe, Germany
**3** Institut Quantique and Département de Physique, Université de Sherbrooke, Sherbrooke J1K 2R1 QC, Canada
* Dietmar.Weinmann@ipcms.unistra.fr

February 16, 2021

## Abstract

**We demonstrate the emergence of classical features in electronic quantum transport for the scanning gate microscopy response in a cavity defined by a quantum point contact and a micron-sized circular reflector. The branches in electronic flow characteristic of a quantum point contact opening on a two-dimensional electron gas with weak disorder are folded by the reflector, yielding a complex spatial pattern. Considering the deflection of classical trajectories by the scanning gate tip allows to establish simple relationships of the scanning pattern, which are in agreement with recent experimental findings.**

# 1 Introduction

Electronic quantum transport in high-mobility two-dimensional electron gases (2DEGs) has been investigated in great detail during the last decades. An important tool to study the transport properties are scanning gate microscopy (SGM) experiments [1–4], which measure the conductance change of a two-dimensional structure that is induced by a local scatterer (usually a charged atomic force microscope tip above the sample). The dependence of such an SGM response on position and strength of the scatterer yields a rich amount of data that contain additional information with respect to that of a standard transport measurement. Particularly prominent quasi-one-dimensional filamentary structures, dubbed "branches", appear in the SGM data at a certain distance from a quantum point contact (QPC) [2,3,5,6]. These branches have been interpreted in terms of inhomogeneous electron flow within the 2DEG, based on the analysis of quantum simulations and classical trajectory density counting performed in the presence of a weakly disordered potential [2,3].[1] While recent works have focused on the stability of these branches when the Fermi energy is varied [6,8], and very recently branches have been observed in light propagation through soap films [9], confining gates have been found to lead to more complex SGM patterns [10–15] which are often difficult to interpret.

When a channel defined by gates beyond the QPC is progressively turned on, three experimental regimes are observed: one in which branches spread unrestrictedly, one in which branches are confined, and one where the branches have disappeared [13]. The case of a circular mirror facing a QPC has been explored experimentally and theoretically [14] as a function of the invasiveness of the tip and the degree of confinement. The length scale of the fluctuations in the SGM response attains a maximum value, which is related to the spatial extension of the tip potential, for the weakest tip strength where the tip can be treated perturbatively [16,17]. As a consequence, the resolution of the SGM response is rather poor in the particularly interesting weakly invasive limit that allows to draw conclusions about the sample under study that are unaffected by the presence of the tip.

When going from the weakly to the strongly confined regime, the gradual loss of a clear branch structure is associated with a folding of the branches, where the SGM pattern resembles a thicket. It is therefore important to determine the information that can be extracted from the effect of an SGM tip in a confined cavity. This is the goal of our work, where we investigate the effect of a micron-sized circular mirror gate facing the QPC on the SGM response as in Ref. [14]. In a recent experiment [18], the SGM response of such a system has been measured as a function of the tip position on a line parallel to the QPC gates and close to the mirror gate as in Ref. [14], and the tip voltage was varied at each position (see Fig. 1). Interestingly, curved lines appear in a plot of the conductance as a function of tip strength and position. On those lines, the conductance assumes particularly high or low values. The present paper

---

[1]It is important to remark that the quantitative description of the branching phenomenon is limited by the lack of a satisfactory definition of what it means to be in a branch (as opposed to not being in one), as well as by the obvious fact that not all the contributing trajectories belong to a branch [7,8].

explains the origin of those structures, and shows that they are related to branches in electron flow through the cavity.

Comparing quantum simulations and classical trajectory approaches, we find that the semiclassical ballistic conductance [19,20] describes well the qualitative features of the transport properties of the system. In particular, we show that the SGM response is related in a subtle way to branches that appear in the electron flow, and which can be deflected by the effect of the tip potential. Such a tip-induced modification of a branch can direct it back into the QPC and reduce the conductance or perturb a branch that is reflected by the mirror into the QPC and thereby increase the conductance. Such an effect on a branch depends on the tip strength and its position with respect to the branch. Features in the SGM response that appear due to this mechanism persist when the tip strength and the tip position vary simultaneously in a way that lets the deflection of the branch constant. Following these features to the limit of weak tip strength allows, in principle, to determine the branch position in the absence of the SGM tip.

Our paper is organized as follows: In Sec. 2, we present quantum and classical approaches to the branching phenomenon in transport through a cavity. The conductance through the cavity and the effect of an SGM tip is addressed in Sec. 3, with particular attention on the relation between the presence of branches and the SGM response. A comprehensive analytical approach to the dependence of the conductance on tip strength and position is presented in Sec. 4, confirming the existence of branch-induced features in the SGM response and the possibility of detecting branches in an unperturbed cavity by SGM experiments. Conclusions are presented in Sec. 5.

## 2 Branches and partial local density of states in open space and in a cavity

In this section, we discuss the effect of a cavity (see the sketch in Fig. 1) on the branches that appear in the electronic scattering wave functions and in the corresponding partial local density of states (PLDOS). The appearance of branches in SGM of electronic transport in a high-mobility 2DEG behind a constriction like a QPC is well known [2,3,5,6]. Its appearance is due to the unavoidable weak disorder seen by the electrons, and the precise branch pattern depends on the details of the disorder realization in the sample, but they are quite stable with respect to changes of the Fermi energy [6,8]. The branches in the SGM response to an invasive tip have been related [21] to the PLDOS for electrons entering from the QPC at the Fermi energy. In general, the PLDOS for electrons with energy $E$ from electrode $l$ with $N_l$ open channels can be defined as [22,23]

$$\rho_{lE}(\vec{r}) = 2\pi \sum_{a=1}^{N_l} |\psi_{lEa}(\vec{r})|^2 \qquad (1)$$

through the scattering wave functions $\psi_{lEa}$ injected from the channel $a$ of electrode $l$. It has been shown that the branching structure is well reproduced by the density of classical trajectories starting in the QPC [2,3,6]. Here we are interested in the consequences of imposing a circular gate facing the QPC, as they pertain to the branch structure and branch folding, as in the sample measured in Ref. [14]. We thus start our analysis by considering the effect of confinement on the PLDOS and on the classical trajectory density.

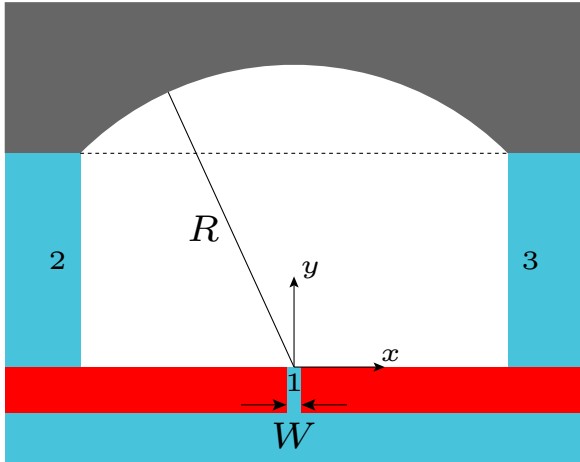

Figure 1: Sketch of the system geometry considered. Red areas represent high potential regions that define the QPC of width $W$. The gray area indicates the mirror potential with a circular edge of radius $R$ that is intended to reflect electrons back into the QPC, and which is present in most of our calculations. In our transport calculations, we consider the conductance between the electrodes 1, 2, and 3 shown in blue, and in particular from electrode 1 to 2 or 1 to 3. The dashed line indicates the possible tip positions that are assumed in our simulations of SGM.

## 2.1 Branches and scattering wave functions

We consider a 2DEG with weak disorder. The disorder potential is generated following the approach of Ref. [24] with realistic parameters (150000 impurities in a square of size $10\,\mu$m $\times$ $10\,\mu$m, in the doping layer situated at a distance of $70\,$nm) as described in Appendix A of Ref. [8]. We first calculate the PLDOS corresponding to scattering states entering the 2DEG at an energy $E = 5.3\,$meV from a QPC and in the absence of the tip, corresponding to a de Broglie wavelength of $\lambda = 65.5\,$nm. The width of the QPC is $W = 100\,$nm, such that three channels are open. The PLDOS obtained without the confining mirror (gray region in Fig. 1) using the fully coherent quantum transport approach implemented through the Kwant package [25] is shown in the left panel of Fig. 2. Branch-like structures are clearly visible in this PLDOS pattern.

In a second calculation, we include the mirror gate (gray region in Fig. 1) at a distance of $R = 2\,\mu$m from the QPC by adding hard wall boundaries. The resulting PLDOS is shown in the right panel of Fig. 2. The first obvious effect of the mirror is that the scattering wave function does not extend beyond the mirror boundary, and only the values inside the cavity are calculated and plotted. Furthermore, one can still notice branches, and some branch folding, but the structure has become richer and more complex. In addition, oscillations appear on the scale of $\lambda/2$ due to interferences of wave-like elements with the reflection from the boundary. These interferences lead to measurable effects when the radius of the mirror cavity is varied [14, 26, 27].

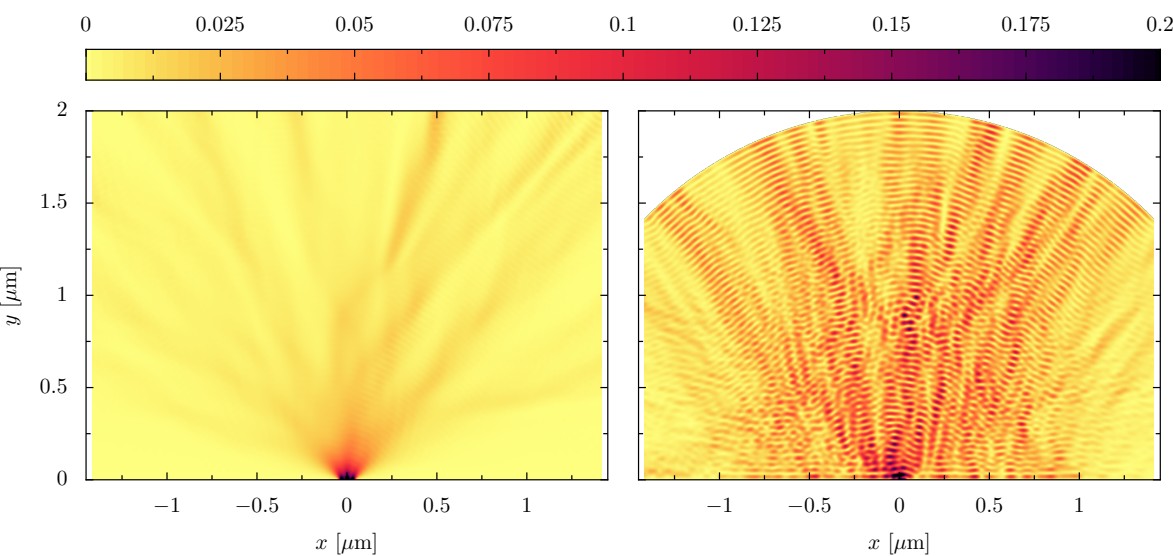

Figure 2: Partial local density of states from the QPC $\rho_{1E}(x, y)$ at an energy $E = 5.3\,\text{meV}$ [see Eq. (1)] for a weakly disordered 2DEG without (left panel) and with (right panel) a cavity mirror.

## 2.2 Description by classical trajectory density

As the main features of the branching phenomenon [2, 3, 6, 8] and the conductance through micron-sized cavities [28] are well described by classical trajectory approaches, we compute the classical trajectories from the QPC and their density as described in Ref. [8]. Figure 3 shows the result for the classical trajectory density in the same system and for the same parameters and disorder configuration as the quantum calculation of the scattering states discussed above. Interestingly, most of the structures in Fig. 2 are reproduced by the trajectory density shown in Fig. 3, except for the interferences described at the end of the previous section.

Comparing the branch structures without and with the confinement due to the mirror, shown in the left and right panel of Fig. 3, respectively, the branches in the confined case include the ones seen in the unconfined situation. After all, the trajectories starting from the QPC are exactly the same until they reach the mirror gate, and therefore the appearing branches are also the same. In the confined case, the trajectories are reflected by the circular mirror, and thus additional structures due to the folded branches appear in the right panel of Fig. 3. The trajectories are typically reflected several times by the mirror and the QPC-forming potential walls until they leave the cavity either through the QPC or on one of the sides. As a consequence, the average trajectory density in the cavity is increased, very much like the PLDOS in Fig. 2. This leads to much richer structures, and it allows to increase the SGM response such that the weakly invasive regime can be investigated experimentally without suffering from a too strong reduction of the signal [14].

## 3 Electronic conductance through a cavity

After having verified that the correspondence between the PLDOS and the trajectory density is approximately maintained upon imposing the confinement, we now turn to the electronic

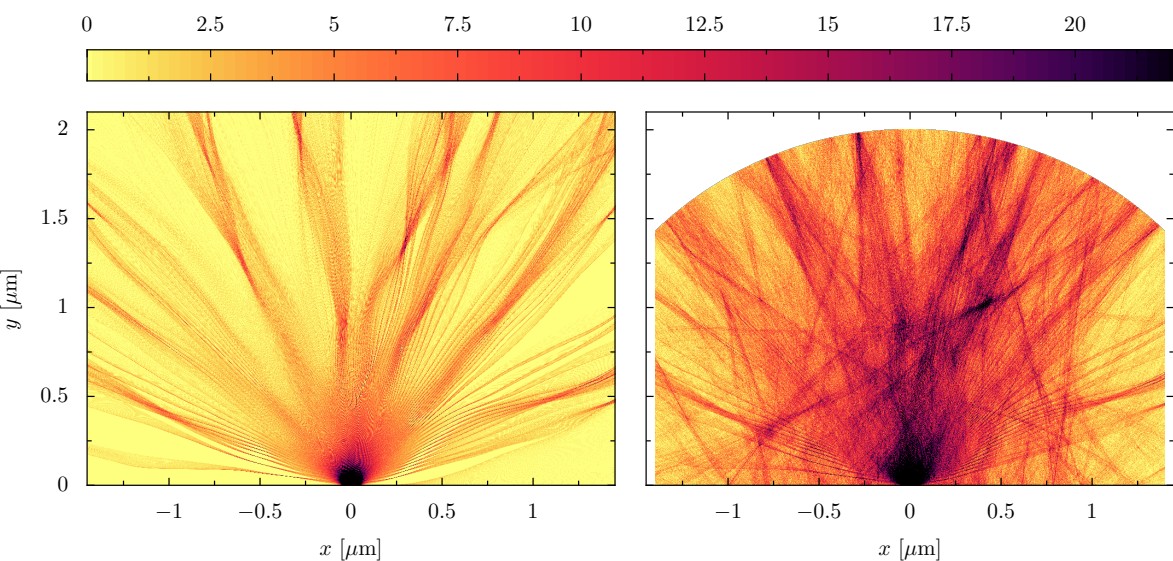

Figure 3: Density of classical trajectories for the same systems and disorder configuration as in Fig. 2. Left (right) panel: without (with) the cavity mirror.

conductance through the device in the presence of the mirror, that is the physical quantity to which we have access experimentally. We consider the conductance between the electrode 1 below the QPC and electrodes 2 and 3 of the cavity (cf. Fig. 1). The latter electrodes are treated as one large reservoir at a common chemical potential, and the conductance we are interested in is given by the sum $G = G_{12} + G_{13}$ of the conductances towards those electrodes.

## 3.1 Semiclassical approach to the conductance of an open cavity

The classical conductance of a ballistic system can be derived using a semiclassical approach [19,20], yielding a weighted average over the contribution associated with transmitted classical trajectories with different initial conditions in the injection lead. In our case, we take the positions $x \in [-W/2, +W/2]$ and $y = 0$ at the interface between electrode 1 and the cavity, and initial angles $\phi \in [-\pi/2, +\pi/2]$ with the $y$ axis (see Fig. 1), to get

$$G = \frac{2e^2 m v_{\mathrm{F}}}{h^2} \int_{-\pi/2}^{+\pi/2} \mathrm{d}\phi \, \cos\phi \int_{-W/2}^{+W/2} \mathrm{d}x \, f(x, \phi), \tag{2}$$

where $e$ is the elementary charge, $m$ the (effective) electron mass, $v_{\mathrm{F}}$ the Fermi velocity, and $h$ is Planck's constant. In the expression above, $f(x, \phi) = 1$ (0) for transmitted (reflected) trajectories as a function of the initial conditions $(x, \phi)$. When a trajectory enters an electrode it is assumed that it ends there and does not come back into the cavity, i.e., a trajectory that ends in regions 2 or 3 of Fig. 1 is counted as transmitted when it reaches one of the two electrodes, and as reflected when it returns to the injection electrode 1. Such a conductance evaluation based on classical trajectories does not reproduce quantum effects like, for instance, ballistic conductance fluctuations. However, as shown in Ref. [28] for similar parameters as those used in this work, a finite temperature, of the order of the one present in the experiment of Ref. [18], leads to a smoothing of the fluctuations, and the resulting finite temperature quantum conductance is well described by the classical conductance (2).

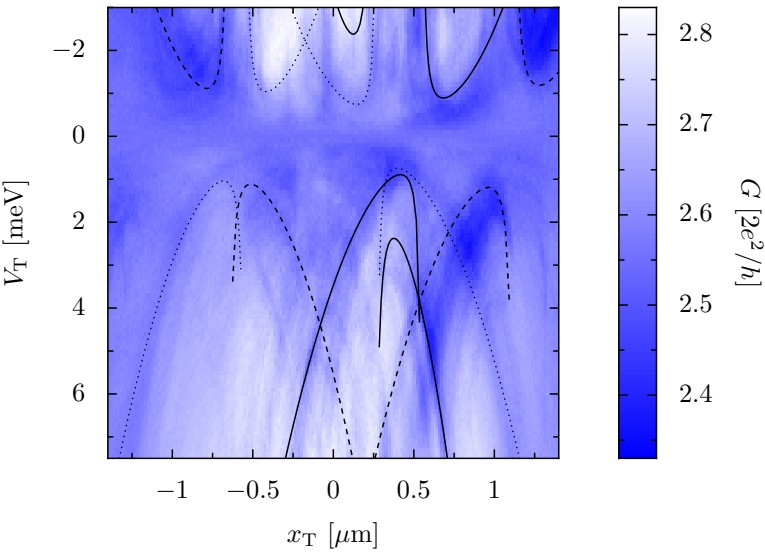

Figure 4: Classical ballistic conductance $G$ from Eq. (2) (in units of the conductance quantum $2e^2/h$) as a function of position $x_T$ (while $y_T = 1.414\,\mu$m is kept fixed along the dashed line in Fig. 1) and strength $V_T$ of the tip. The various lines represent plots of Eq. (7).

The conductance $G$ obtained using a numerical implementation [28] of Eq. (2) for the same disorder realization as in Sec. 2, and with a hard-wall mirror cavity, is about $G \simeq 2.6 \times 2e^2/h$, thus not far from the transmission $T = 3$ of a QPC on the third conductance plateau. Such a value indicates that in spite of the perfect reflection of the mirror, most of the electrons injected into the cavity from the QPC are not directed back into the QPC. This phenomenon illustrates the crucial role that has the small-angle scattering due to weak and smooth disorder [14].

### 3.2  Conductance in the presence of an SGM tip

We now focus on the SGM response (the conductance change induced by a local potential) as a function of tip strength and position, and the relation of its features to the branches in the cavity. In the presence of an SGM tip potential of Lorentzian shape

$$V(\vec{r}) = V_T \, \frac{\sigma^2}{|\vec{r} - \vec{r}_T|^2 + \sigma^2} \tag{3}$$

with an amplitude $V_T$ that determines the tip strength and a realistic [14] width $\sigma = 175\,$nm, the conductance can be increased or reduced, depending on the details of the tip parameters and its position $\vec{r}_T$ related to the electron flow in the cavity. The calculated conductance as a function of the position $x_T$ (on the dashed line in Fig. 1 with $y_T = 1.414\,\mu$m fixed) and strength $V_T$ of the tip is presented in Fig. 4.

Figure 4 exhibits different kinds of structures in the SGM response. We have checked that they are similar to the ones seen in the quantum conductance (see Ref. [18] and Appendix A), indicating that the semiclassical approach represents a rather good approximation. The observations are also consistent with the experimental data of Ref. [18].[2]  Among them, a

---

[2]The relation between the voltage applied to the tip in the experiment and the maximum tip potential $V_T$ is approximately linear with a tip voltage of about $-1\,$V leading to $V_T = 1\,$meV [14].

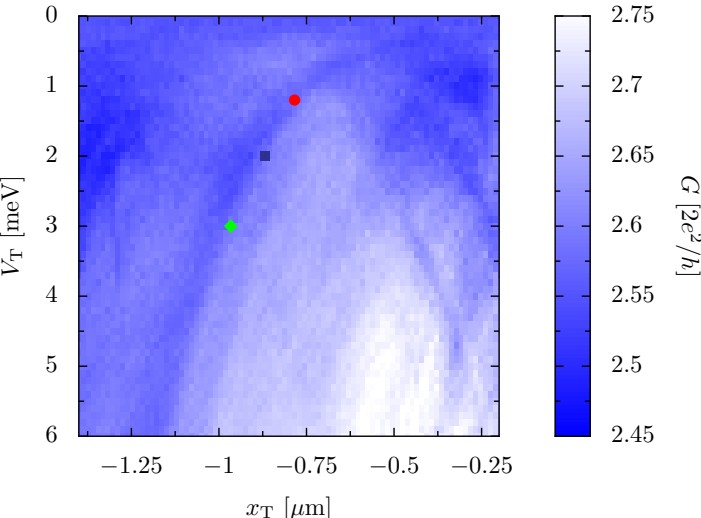

Figure 5: Zoom in the lower left part of Fig. 4. The three marked points on the prominent edge between high and low conductance represent the parameter values for which the classical trajectory density is shown in Fig. 6.

strong repulsive ($V_\text{T} > 0$) or attractive ($V_\text{T} < 0$) tip close to the center of the cavity increases the scattering of the electrons towards the side electrodes 2 and 3, and thereby increases the conductance to values that approach the transmission $T = 3$ of the QPC.

More intriguing, there are lines with approximately constant conductance that appear in Fig. 4 between regions of strong and weak tips, and which have a characteristic parabola-like shape. Such structures exist in the upper part ($V_\text{T} < 0$) where the tip potential is attractive as well as in the lower part ($V_\text{T} > 0$) where it is repulsive. Several of those structures seem to be superposed. We will argue in the sequel that a classical mechanism of bending of the branches by the effect of the tip is at the origin of these structures in the conductance plots.

In Appendix B we tackle the semiclassical computation of the conductance for a cavity without disorder (i.e., the clean case) in the presence of an SGM tip, where an analytical treatment is possible for particular shapes of the tip potential. Contrary to the previously discussed case of a disordered cavity, the redirection into the QPC by the mirror is very efficient, and the conductance for weak values of the tip strength is very low. Despite this important difference, the parabolic-like features present in the experiment [18] and reproduced by the semiclassical approach leading to Fig. 4, are also suggested in the clean case (see Fig. 12).

## 3.3 Signature of branches in the SGM response

In order to illustrate the above-mentioned mechanism, here we focus on one of the prominent curved equiconductance lines in the position-amplitude space of Fig. 4, and calculate the tip effect on the classical trajectory density for three points on that feature. A zoom of the selected region with three colored markers that represent the chosen points in a prominent structure is shown in Fig. 5. The classical trajectory density in the presence of the tip for the three parameter sets is shown in Fig. 6. It can be seen that the structure of the branches in the cavity does not change drastically under the effect of the tip, and that the branches

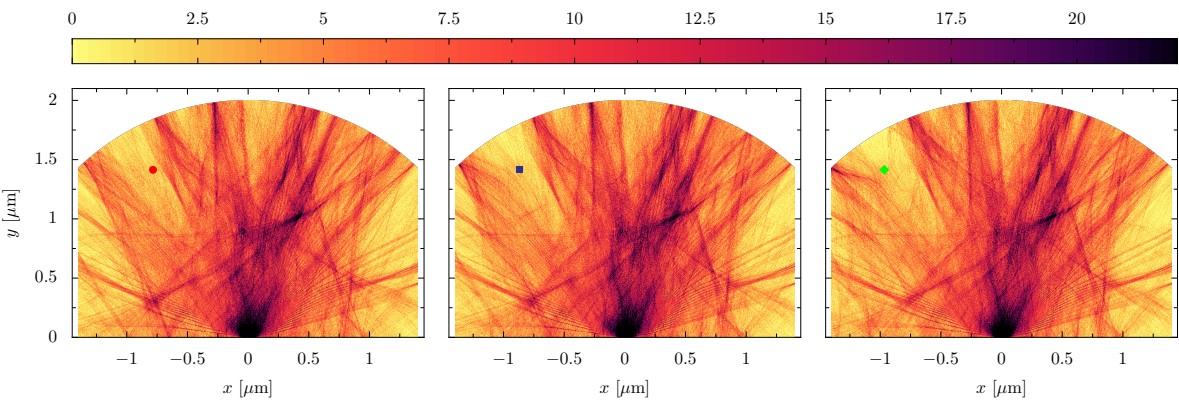

Figure 6: Density of classical trajectories in the weakly disordered sample of Fig. 2, modified by a Lorentzian bump at three different positions and strengths on the feature shown in Fig. 5. The position of the tip is indicated by the colored symbols in the panels, that allow to identify the parameters with the ones of the marked points in Fig. 5.

close to the tip are deflected by the tip potential in a way that keeps their contribution to the conductance essentially unchanged. For the most significant and closest branch, the change in tip position is compensated by a change in tip strength: A strong tip that is far away may have a similar effect as a tip that is weak and close.

Based on such a qualitative understanding of the equiconductance lines, we propose in the next section a model calculation that leads to a detailed quantitative description of the shape of these lines.

## 4    Analytical model describing equiconductance features

In this section, we present a model calculation to predict the form of the equiconductance lines in SGM measurements as a function of tip position $x_T$ and strength $V_T$. The model is based of the deflection of classical trajectories by the tip potential, and the assumption that the tip effect on the contribution of a branch to the conductance is determined by the deflection angle of the trajectories in the branch.

Since the classical conductance (2) measured in the SGM setup crucially depends on whether trajectories are transmitted or not, we expect that the SGM response is dominated by a number of branches that form due to weak disorder [8], and whose transmission is modified by the effect of the tip potential. Branches are bundles of trajectories, and thus the impact of the tip can be approximately described by the deflection of a trajectory that is part of the branch.

For the quantitative description of such a deflection, we assume that the tip-induced potential bump/dip has circular symmetry around the tip center $\vec{r}_T$ with the general expression

$$V(\vec{r}) = V_T \, v(|\vec{r} - \vec{r}_T|), \tag{4}$$

where $V_T$ is the potential strength, and $v(r)$ the shape of the potential with normalized maximum value $v(0) = 1$ at the tip center. In the limit of weak scattering, the deflection angle $\alpha$ due to the presence of the potential (4) for a trajectory at energy $E$, starting far from

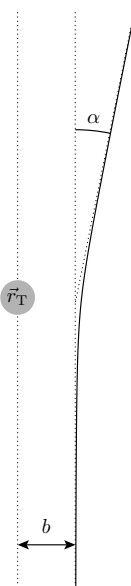

Figure 7: Sketch of the considered situation in the calculation of the deflection of a classical trajectory.

the tip position $\vec{r}_{\mathrm{T}}$ and missing the tip by the impact parameter $b$ (see the sketch in Fig. 7), can be calculated [8] as

$$\alpha = -\frac{V_{\mathrm{T}}}{E} b \int_1^\infty \mathrm{d}s \, \frac{v'(|b|s)}{\sqrt{s^2 - 1}}, \tag{5}$$

where $v'$ is the derivative of $v$ with respect to $r$.

The effect of the tip described by the deflection angle (5) depends on the energy, shape of the tip potential, tip position, and tip strength. For the experimentally realized situation of fixed energy and tip potential shape, the condition of a constant (small) deflection angle for a given branch leads to a relationship between $b$ and the corresponding $V_{\mathrm{T}}^*$, by recasting Eq. (5) as

$$V_{\mathrm{T}}^* = -\alpha E \left[ b \int_1^\infty \mathrm{d}s \, \frac{v'(|b|s)}{\sqrt{s^2 - 1}} \right]^{-1}. \tag{6}$$

As the contribution of a tip-deflected branch (represented by a single trajectory) to the SGM response remains approximately unchanged along lines described by the relation (6), prominent features in the SGM response can be expected to appear on such lines for particular values of $\alpha$, for example the ones for which an otherwise transmitted branch is reflected back into the QPC.

Evaluating Eq. (6) for a Lorentzian potential bump/dip $v(r) = \sigma^2/(r^2 + \sigma^2)$ of width $\sigma$, we obtain the result

$$\frac{V_{\mathrm{T}}^*}{E\alpha} = \frac{2}{\pi} \frac{\left[ (b/\sigma)^2 + 1 \right]^{3/2}}{b/\sigma} \tag{7}$$

for the expected shape of SGM features related to a constant deflection angle $\alpha$, in terms of the distance $b$ between the branch and the tip. The result of Eq. (7) is depicted by the solid lines in Fig. 8. Interestingly, the behavior for tip positions far from the main branch when

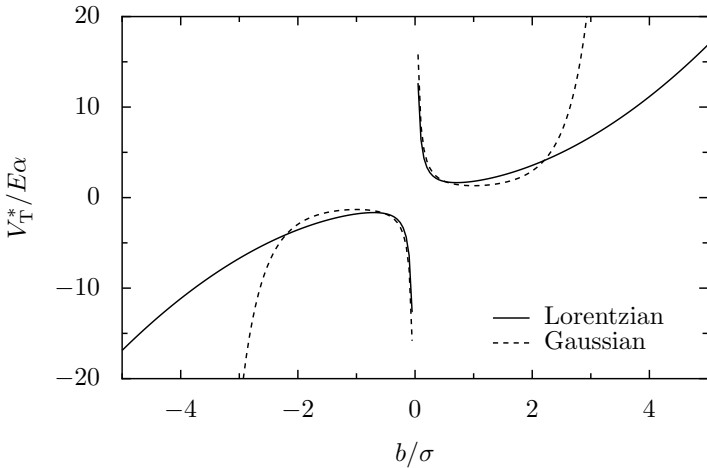

Figure 8:   Plot of the tip strength needed to maintain constant deflection as a function of distance from the branch for Lorentzian [solid lines, Eq. (7)] and Gaussian [dashed lines, Eq. (9)] tip potential shapes.

$b \gg \sigma$ given by

$$\frac{V_{\mathrm{T}}^*}{E\alpha} \simeq \frac{2}{\pi}\left(\frac{b}{\sigma}\right)^2 \tag{8}$$

is approximately parabolic, consistent with the experiment [18].

For a radial trajectory starting at the origin with an angle $\phi$ to the vertical ($y$) axis, the impact parameter depends on the tip position as $b = (x_{\mathrm{B}} - x_{\mathrm{T}})\cos\phi$, with $x_{\mathrm{B}} = y_{\mathrm{T}}\tan\phi$ the $x$-position where the branch crosses the horizontal line $y = y_{\mathrm{T}}$ on which the tip is moving in the experiment of Ref. [18]. The relation between $b$ and $x_{\mathrm{T}}$ allows us to obtain the shape of the features in the $x_{\mathrm{T}}$–$V_{\mathrm{T}}$ plane from Eq. (7) through an offset $x_{\mathrm{B}}$ and the scale factor $\cos\phi$. The lines in Fig. 4 are examples of the resulting predicted shape of constant deflection angle features, showing that the proposed mechanism is consistent with the structures appearing in the classically calculated conductance. Solid, dashed, and dotted lines, respectively, correspond to values of branch positions and deflection angles $(x_{\mathrm{B}}, \alpha)$ of $(0.55, 0.018)$, $(1.12, 0.024)$, and $(-0.55, 0.021)$ for positive deflection angles, and $(0.25, -0.047)$, $(-0.65, -0.022)$, and $(0.27, -0.015)$ for negative deflection.

In order to investigate the influence of the tip-induced potential profile on the expected shape of equiconductance lines in the SGM response, we also present the case of a Gaussian potential bump/dip shape $v(r) = \exp\left(-r^2/2\sigma^2\right)$, for which one obtains from Eq. (6) the behavior

$$\frac{V_{\mathrm{T}}^*}{E\alpha} = \sqrt{\frac{2}{\pi}}\frac{\exp\left([b/\sigma]^2/2\right)}{b/\sigma}. \tag{9}$$

Equation (9) is shown through the dashed lines in Fig. 8. Their shape agrees with the features found in the quantum transport simulations of Appendix A, where a Gaussian tip potential shape has been used. This is demonstrated by the dashed lines in Fig. 11 that represent feature shapes resulting from Eq. (9).

It can be seen from Eq. (6) and the examples plotted in Fig. 8 that the form of the tip-induced potential, and in particular its behavior far from its center, strongly influences the shape of the constant deflection angle features. Therefore, a comparison of our results with

the experimentally observed features could be used in order to determine the tip-induced potential profile. The fact that the features in the experiment of Ref. [18] are approximately parabolic allows us to conclude that the tip-potential is close to a Lorentzian form, consistent with the results of other studies using different methods. Examples are Ref. [29], where the tip-induced potential is inferred from its effect on the conductance of a nearby QPC, and Ref. [30], where the size of the depletion disk of a strongly invasive tip is extracted from Fabry-Perot interference fringes.

A common feature of both, Lorentzian and Gaussian tip-induced potentials, is the divergence $V_T^* \propto \sigma/b$ in the limit $b/\sigma \to 0$, when the tip center approaches the trajectory. This divergence is due to the flat summits of the tip potentials considered, and the resulting need of stronger and stronger tip potentials to maintain a constant deflection angle. However, in this limit, trajectories are blocked when $V_T^* > E$, and we do not expect strong features at very large tip strength in the experiment. Anyway, due to the symmetry of the potential, no deflection is possible when $b = 0$, and it is clear that the lines of continuous deflection in Fig. 8 must have two non-connected parts. One of them corresponds to a repulsive tip on one side of the branch, the other to an attractive tip on the other side. The two are related by the odd symmetry of the features $V_T^*(-b) = -V_T^*(b)$, reflecting the fact that a fixed deflection can be due to pushing from one side (positive $V_T$) or pulling from the other side (negative $V_T$).

While the sketch of Fig. 7 and the relationship (5) seem to apply to a trajectory belonging to a bundle that emerges from the QPC, the same kind of reasoning can be used in the case of a trajectory that has bounced off the reflector, and thus belongs to a folded branch. It is worth to mention that Eq. (5) is valid for any kind of weakly deflecting potential bump or dip. In this work we have applied it to the case of the deflection induced by the tip onto a classical trajectory in the absence of disorder, while in Ref. [8] it was used to understand the caustic formation due to weak disorder. In the realistic case of a disordered cavity, $\alpha$ would describe the additional deflection induced by the tip in an almost straight trajectory, and the effect of a weak disorder does not modify the simple picture arising from the analytical model.

The described tip effect leading to conductance features for particular constant deflection angles is expected to occur in more general situations that are not restricted to the particular cavity geometry considered in our numerics. For the effect to occur, two ingredients are needed. First, weak disorder leading to branch formation on the scale of the cavity should be present. Second, one or more of the contacts used to measure the conductance should be small enough to resolve the contribution of individual branches being reflected or transmitted. In contrast, the precise shape of the cavity should not be of crucial importance. For example, in the presence of weak disorder, a mirror with an imperfect almost circular shape should lead to a behavior that does not change qualitatively as compared to the one obtained with a perfectly circular mirror.

## 5 Conclusions

Motivated by recent experimental results [18], we have performed quantum and classical numerical simulations of transport through a QPC that is facing a circular reflecting mirror gate, taking into account the effect of a perturbing tip voltage. The numerics confirmed that an approach based on classical trajectories is able to capture the main features of the exper-

imentally observed SGM patterns, and in particular it reproduces parabola-shaped features in the conductance plots as a function of tip position and strength. From analytical model calculations, we reach an understanding of the underlying mechanism, which is based on the branching of electron flow in a weak disorder potential, and considers the effect of the tip potential on nearby branches. A simple model describing the scattering of classical trajectories allows to predict features of SGM in the $x_{\mathrm{T}}$–$V_{\mathrm{T}}$ plane where the deflection angle of a branch due to the effect of the tip remains constant. When the tip is moved away from the branch, its effect on the corresponding electron trajectories gets weaker, while with a simultaneous increase of tip strength, the influence on the trajectories can remain almost unchanged, giving rise to the observed "parabolic" features. All these predictions are in good agreement with experiments [18] and simulations of the SGM response, including the fact that the features exhibit a gap between the regimes of repulsive and attractive tips. It follows that the analysis of features of the SGM response as a function of tip position and strength can be used to extract the positions of branches in the unperturbed system (without the tip).

In addition, the feature geometry depends on the shape of the potential of the bump or dip, with almost parabolic features for a Lorentzian tip potential and large $|b|/\sigma$. Notably, such a dependence could allow to determine experimentally the shape of the tip potential from a comparison of the observed constant deflection-angle features with the shape-dependent prediction of Eq. (6), and provide data that are independent of other suggested methods [29, 30]. The form of the observed features in the SGM response of a 2DEG [18] is consistent with a Lorentzian shape of the tip potential.

The classical limit of the semiclassical approach has been found to be powerful enough to explain the experimental data for 2DEGs and the quantum simulations, unlike the case of open microwave billiards, where the influence of diffraction has been put in evidence [31, 32].

We have demonstrated that, in the regime of strong confinement, important information can be obtained by analyzing the features of the folded-branch thicket-like SGM pattern. While weak disorder is crucial for the establishment of the branches and for certain observables like the conductance of a mirror-confined cavity, it does not prevent the emergence of classical trajectory features underlying the SGM pattern.

## Acknowledgements

We are indebted to Carolin Gold, Klaus Ensslin, and Thomas Ihn for sharing with us unpublished experimental results which motivated the present work. We are grateful to Nicolas Cartier for useful discussions.

**Funding information**   Financial support from the French National Research Agency ANR through projects ANR-11-LABX-0058_NIE (Labex NIE) and ANR-14-CE36-0007-01 (SGM-Bal) is gratefully acknowledged. This research project has been supported by the University of Strasbourg IdEx program.

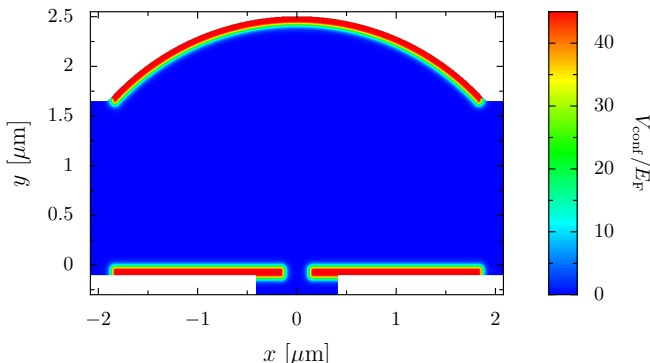

Figure 9: Confinement potential $V_{\mathrm{conf}}$ (in units of the Fermi energy $E_{\mathrm{F}}$) with smooth walls defining the geometry of Fig. 1 used for the quantum conductance calculations. White areas are not part of the system, equivalent to regions of infinite potential.

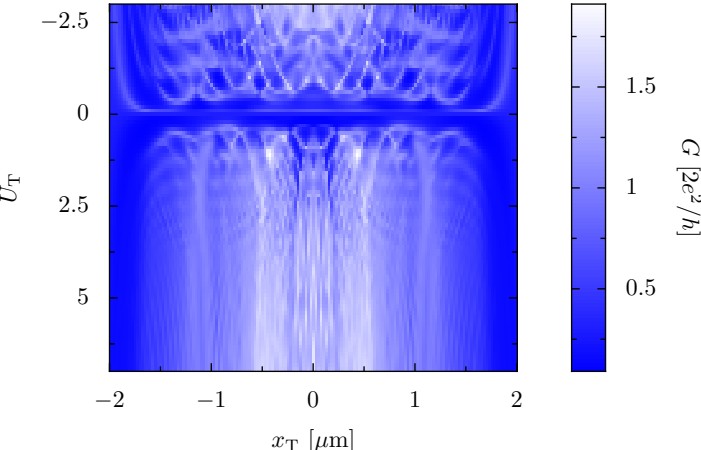

Figure 10: Quantum conductance (in colorscale) through the model cavity in the absence of disorder, as a function of tip position $x_{\mathrm{T}}$ at fixed $y_{\mathrm{T}} = 1.625\,\mu\mathrm{m}$, and (dimensionless) tip strength $U_{\mathrm{T}}$.

## A    Quantum simulation of SGM in a cavity

We have performed quantum transport simulations using the Kwant package [25]. In these simulations, we calculate the electronic transport properties assuming that the electrons move in an effective one-body potential that determines the cavity as shown in Fig. 9, and that electron-electron interaction effects beyond mean field can be neglected. The non-abrupt character of the chosen confining potential diminishes the importance of diffraction effects [31,32] appearing in this kind of geometry. The electron dynamics is assumed to be perfectly coherent, and the transport properties are calculated at zero temperature. The parameters of the cavity shape and its size as well as the Fermi wavelength are close to the experimental situation of Ref. [18]. For the tip potential, a Gaussian shape with strength $U_{\mathrm{T}}$ similar to the one used in Ref. [14] is used. Roughly, $U_{\mathrm{T}} = 1$ corresponds to an experimental tip voltage of $-1\,\mathrm{V}$. Similar to the experiment, the tip positions are chosen to vary along a line at $y_{\mathrm{T}} = 1.625\,\mu\mathrm{m}$.

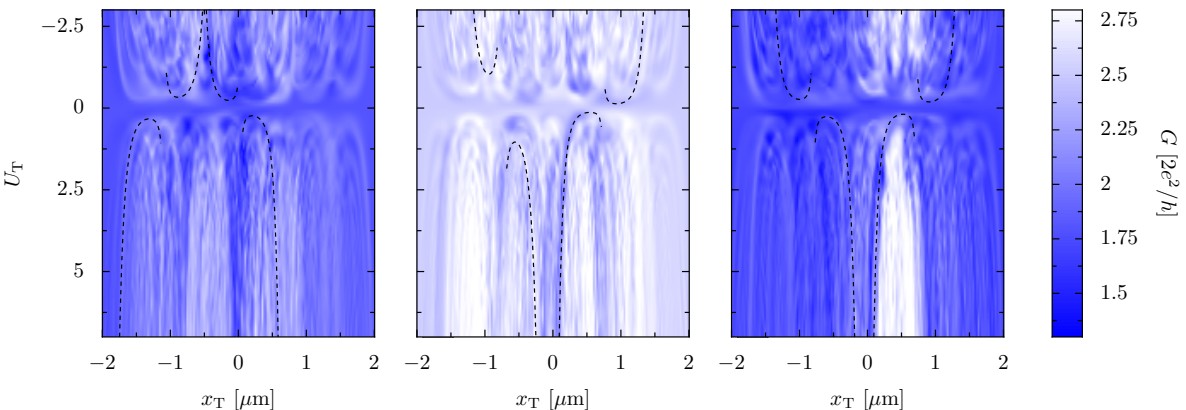

Figure 11: Quantum conductance through three samples with different disorder realizations. The consistence with our deflection mechanism of Sec. 4 is shown by the lines that represent constant deflection angle features according to Eq. (9). The parameters $(x_B, \alpha)$ are $\{(-1.1, 0.0083), (0.03, -0.0059)\}$, $\{(0.74, 0.0033), (-0.75, -0.026)\}$, and $\{(0.715, 0.0047), (-0.8, -0.0066)\}$ for the left, central, and right panel, respectively.

The numerical result for the quantum conductance in a clean model is shown in Fig. 10 as a function of tip position and strength. In the absence of disorder, and without the tip, the cavity reflects most of the electrons back into the QPC, such that the conductance at $U_T = 0$ is quite low. The main consequence of the tip is to perturb this mirror effect and to enhance the conductance. However, as in the experiment, the conductance change due to the tip exhibits the characteristic features evident in the plot of Fig. 10.

The situation changes in the presence of weak disorder, where already in the absence of the tip, most electrons are transmitted, such that the conductance in this case is strongly enhanced with respect to the clean case [14]. We use a realistic disorder strength as in Ref. [14], implemented along the lines described in Ref. [24] and detailed in Appendix A of Ref. [8]. The results for the conductance as a function of tip position and strength are shown for three different disorder realizations in Fig. 11. It can be observed that there are curved features in all plots, some of them where the conductance is enhanced and some with reduced conductance. Their positions depend on the disorder configuration.

The form of the features is well described by the constant deflection angle approach of Sec. 4, as demonstrated by the dashed lines in Fig. 11. Those lines are plots of Eq. (9) which is derived for a Gaussian tip potential as it is used for the quantum transport calculations of this Appendix. Quantum simulations with a Lorentzian tip potential [18] lead to arc-like features of approximately parabolic form comparable to our classical transport results of Sec. 3 and the model prediction (7).

## B    Classical ballistic conductance through a clean cavity

In this Appendix, we present an analytically solvable model that describes the conductance through a clean cavity sample in the presence of a perturbing tip potential. As in Ref. [28], we evaluate the conductance (2) by counting transmitted and reflected trajectories from an ensemble of initial conditions in the injecting lead [19,20]. We consider the case of a disorder-

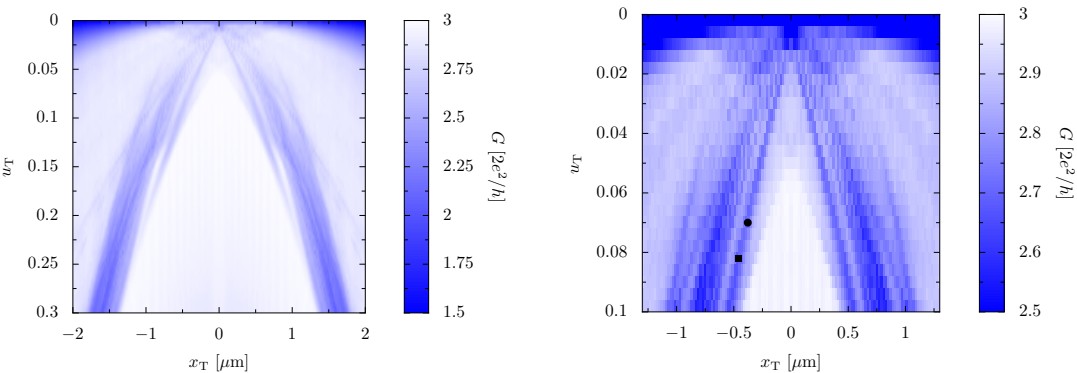

Figure 12: Classical conductance through a clean cavity calculated from exact trajectories in a $1/r^2$-shaped tip potential. The right panel shows a zoom around the origin of the left panel using a more sensitive colorscale in order to highlight the details. The circle and the square indicate parameter choices for which the reflected trajectories are shown in Fig. 13.

free system for which we can analytically calculate the electron trajectories at energy $E$. We also assume that the sample confinement is due to hard-wall potentials with specular reflection of the trajectories, and that the tip potential reads

$$V(\vec{r}) = u_{\mathrm{T}} E_{\mathrm{F}} \left( \frac{R}{|\vec{r} - \vec{r}_{\mathrm{T}}|} \right)^2 \tag{10}$$

with $u_{\mathrm{T}}$ some dimensionless tip strength and $E_{\mathrm{F}}$ the Fermi energy. The tip center is placed on a line parallel to the $x$ axis as in the experiment. It is then possible to calculate analytically the classical trajectories for the case of a repulsive tip, and thus the conductance based on those trajectories using Eq. (2). The unrealistic divergence of the potential (10) $\propto 1/|\vec{r} - \vec{r}_{\mathrm{T}}|^2$ arising when $\vec{r} \to \vec{r}_{\mathrm{T}}$ is irrelevant in the repulsive case since electron trajectories at a fixed energy $E$ can never reach regions where the potential is larger. Moreover, the difference between a Lorentzian potential and the potential (10) becomes negligible when the potential maximum is much higher than $E$. In the case of an attractive potential however, the negative divergence of the potential leads to artifacts, namely trajectories that get caught in the attractive potential, getting closer and closer to the singularity while accelerating to higher and higher velocity. For this reason, the tip-induced potential can only be approximated by the form (10) when the tip potential is repulsive. We therefore do not include the parameter region of attractive tip potentials in this Appendix.

Figure 12 shows the conductance computed from the analytically calculated trajectories when perturbing the system with a tip potential given by Eq. (10). In the absence of the tip potential, the electron trajectories are straight lines reflected at the confining walls, therefore almost all trajectories that hit the mirror gate are reflected back into the QPC. This leads to a very low conductance $G$ at zero tip strength $u_{\mathrm{T}} = 0$, reminiscent of the behavior found in the quantum simulation of the conductance in the absence of disorder presented in Fig. 10. The tip deflects the electron trajectories such that most of the reflected ones are transformed in transmitted trajectories. Hence, the effect of the tip is to strongly enhance the conductance, and the classical image directly explains this aspect found in the quantum simulation of Fig. 10. However, as can be observed in Fig. 12, in the parameter space where the transmission is enhanced (light blue) by the repulsive tip, there are "parabolic" features where significant

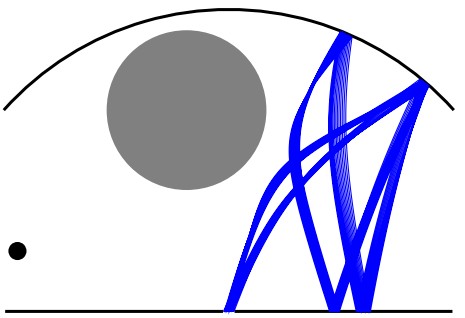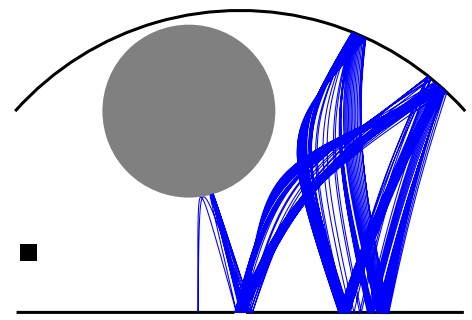

Figure 13: Reflected trajectories for the two marked points of enhanced conductance in Fig. 12. Gray areas: Classically forbidden depletion disks of the tip potential.

reflection is still present (dark blue).

To get a better understanding of the mechanism, we show in Fig. 13 the reflected trajectories for a few parameter values that correspond to the marked points in the right panel of Fig. 12. Two different parameter sets have been chosen on a prominent "parabolic" feature. It can be seen in Fig. 13 that the reflected trajectories for the two cases (one for a tip that is closer to the center and weaker than the other) are very similar and correspond to a family of trajectories that is focused back into the QPC after several reflections at the sample walls. For parameter values placed on other "parabolic" features, the trajectories look quite different, but they also remain similar when one changes parameters along the feature. In contrast to the disordered case, trajectories with only one reflection at the mirror do not play an important role for the equiconductance features in the clean case.

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
