# Peer review of "Signatures of folded branches in the scanning gate microscopy of ballistic electronic cavities"

_SciPost Physics Core_

## Round 2 · Referee Report · Anonymous (Referee 1) · 2020-12-11

Report

In their well-written and comprehensive manuscript Fratus et al. consider scanning-gate probed quasi-ballistic transport through a specific mesoscale cavity in a rather thorough and profound way. In two-dimensional semiconductor cavities smooth disorder gives rise to branched flow that needs to be considered when analyzing the role of a scanning gate on the conductance. The authors give a detailed analysis of the flow, including the reflection from a refocusing circular segment, and its effect on the conductance from a quantum point contact to either of two terminals. When computing the conductance as a function of tip strength and position they find peculiar parabola-shaped structures of nearly constant conductance. As major part of the work and major result they give an analytical explanation of these, at first glance, strange structures and a corresponding picture in terms of branches deflected by the STM tip.

As mentioned the overall analysis is rather profound and scientifically sound. Both KWANT quantum calculations and classical trajectory simulations of the branched flow coincide very well. The results ar relevant in view of the fact that they are directly applicable to experiment and the specific mechanism may help to quantitatively calibrate the STM’s effective potential. Hence I recommend publication, but have a few questions to the authors:

1.) The analysis of the parabolic conductance structures is compared only to the classical results in Fig. 4, while corresponding quantum results are relegated to the appendix. How good does the result of Eq. (7) coincide with the corresponding quantum conductance traces, say in Fig. 11 (that are eventually relevant for experiment)?

2.) The analysis leading to Eqs. (7,8) appears to be independent of the overall (partly circular) cavity geometry. Hence are the parabolic structures in the conductance also independent of that? This would be surprising.

3.) If instead of the sum $G = G_{12} + G_{13}$ the conductances $G_{12}$ or $G_{13}$ were considered individually, how well do classical and quantum results would coincide? Are there non-local quantum effects to be expected? How relevant are those?
  • validity: -
  • significance: -
  • originality: -
  • clarity: -
  • formatting: -
  • grammar: -

Author:  Dietmar Weinmann  on 2021-02-16  [id 1245]

(in reply to Report 1 on 2020-12-11)

We thank the referee for his/her positive overall judgment and constructive remarks. We have taken into account the suggestions as explained in detail below.

  • Following the suggestion of the referee to compare the analytical results of Sec. 4 not only to classical conductance simulations but also to the quantum simulations of Appendix A, we have introduced in Fig. 11 curves (dashed) indicating the prediction of Eq. (9). Therewith, Fig. 11 shows that the features seen in the quantum calculations are consistent with the proposed mechanism of constant deflection angles. The quantum simulations in the Appendix are performed with a Gaussian tip potential, confirming the importance of the shape of the tip potential that is discussed in more detail in Sec. 4 and in the Conclusions of the revised version.
  • The referee points to the fact that the derivation of Eqs. (7,8) does not use the precise shape of the cavity, and we thank him/her for raising that interesting point. We believe indeed that the shape of the cavity is not important for the mechanism put forward, and we have added a discussion of that issue at the end of Sec. 4.
  • The referee points to the possible importance of nonlocal quantum effects making the conductance of two scatterers connected “in parallel” to departure from the simple addition law of classical physics. In an intuitive semiclassical picture, such a quantum effect arises from the multiple electron traversals of the scatterers, as it would for instance occur between two resistors placed on the two arms of an interferometer. However, the geometrical configuration of the device under consideration prevents multiple scattering processes, since once an electron arrives to the contacts 2 or 3, it leaves the device reaching the corresponding electrode without reinsertion in the system. Therefore, the classical addition law of the conductance applies for G_{12} and G_{13}.

---

## Round 2 · Referee Report · Anonymous (Referee 2) · 2020-12-22

Strengths

This is a well written paper, which addresses a question of direct experimental relevance, and provide a theoretical explanation of one of the observation made in ref [17].

Weaknesses

1) A lot of of comparisons are made at a qualitative level (ie by comparing two figures), and in some instances, the reader would clearly benefit from a more quantitative evaluation of a given approximation/approach.

2) Even if the question of "branches" in scanning gate microscopy has been already discuss in previous papers, it would help the non-specialist reader to have a more formal definition of what is a "branch" (I assume related to the presence of disorder induced caustics in the classical dynamics ). This would help in particular clarifying the mindset behind the approximation scheme of section 4 and understanding how exactly are constructed the curves in Fig 4.

3) The derivation of section 4 assume an initial propagation along the vertical direction when the actual propagation (in the absence of disorder) is radial, and it is not clear how this is taken into account.

4) The experimentally observed parabolic form of the equi-conductance curve of Fig 4 seem to be very dependent of the precise form of the tip potential (as it is obtained for the Lorenzian tip potential shape Eq(7), but not at all with a Gaussian tip potential shape Eq(9). However, very little is concluded from this fact.

5) The status of the appendices is a bit unclear.

Report

This is an interesting paper, but the authors should address the point raised above (and made more precise in the list below) before publication.

Requested changes

1-The statement that "most of the structures in Fig. 2 are reproduced by the trajectory density shown in Fig. 3" should be substantiated by a more quantitative analysis.

2-In the same way, the statement in the second paragraph of sec 3.2. that "[the classical conductances] are similar to the ones seen in the quantum conductance (see Appendix A), indicating that the semiclassical approach represents a rather good approximation" should be substantiated by a more quantitative analysis.

3- The details of how the lines corresponding to Eq(7) have been drawn on Fig 4 should be made explicit. In particular : - How was the origin of the \Delta x chosen ? - How where chosen the particular E\alpha plotted ? - Was the circular geometry taken into account ? Again, having a more in depth discussion of what are trajectory branches could help in the discussion of at least the first of these points.

4- More discussion should be given to the importance of the shape of the tip (especially at long distance from the center of the type). Indeed the Lorentzian shape seem to be necessary to get the observed parabola. We therefore need to know how probable it is that the experimental tip potentials have this shape. And if they have this shape, why are the quantum calculation made with a Gaussian shape ?

5- The role and construction of the appendices should be reconsidered. Indeed, I am used to see appendices meant essentially to provide technical details about a particular numerical or analytical computation or experimental setup. Here however both appendix contains a fair amount of discussions (and not that many technical details).
I would tend to think that either the discussion in an appendix is relevant to the general problem considered, and then it should be included in the main text, or it is considered as not sufficiently important to be inserted in the main text, and then presumably it is best left away from the paper altogether.

  • validity: good
  • significance: good
  • originality: good
  • clarity: high
  • formatting: excellent
  • grammar: excellent

Author:  Dietmar Weinmann  on 2021-02-16  [id 1244]

(in reply to Report 2 on 2020-12-22)

We thank the referee for his/her positive overall judgment and useful remarks. We have taken into account the suggestions as explained in detail below.

  • The referee suggests more quantitative comparisons between the quantum partial local density of states in Fig. 2 and the density of classical trajectories of Fig. 3. However, the quantitative comparison of the two landscapes is a difficult task. Moreover, even if we computed a correlation of the quantities plotted in the figures, such a correlation would still depend on an arbitrary parameter, hindering a quantitative comparison. In response to the criticism 2) of the referee, we now stress in the manuscript the difficulties of quantifying branches and indicate two references discussing this issue. We believe that the presented qualitative comparison shows well the similarity of the branch-like structures obtained in the study of different physical quantities. Another suggestion concerns the comparison of the quantum conductance in Appendix A with the classically computed value of Sec. 3.2. While the conductance without the tip is in good agreement, a quantitative comparison in the presence of the tip is not expected to be conclusive, for the following reasons: Firstly, the shape of the tip-induced potential used in Appendix A is Gaussian and thus different from the one used in Sec. 3.2 of the main text (Lorentzian shape), which allows us to compare the effects of different tip-induced potential profiles. Secondly, the conductance changes with the disorder configuration, as shown by the three different realizations in Fig. 11. Notwithstanding, we have performed a detailed study comparing quantum and classical conductance as a function of tip strength in another cavity geometry (see Ref. [28] of the revised paper). A discussion of this issue has been added in Sec. 3.1.
  • In following the referee's recommendation, in the Introduction we now provide a definition of what is a “branch”. Moreover, we added a footnote as a caveat pointing to the difficulty of quantifying the branching phenomenon, and give two references that discuss this important issue.
  • We thank the referee for his/her remark about the details of the derivation in Sec. 4. The derivation of the deflection from a tip potential is independent of the initial direction and only depends on the impact parameter. To make this point clear, we have relabeled the impact parameter in Sec. 4 from "\Delta x" into "b", and we have made minor changes in the text to indicate that the derived deflection holds for an initially radial trajectory. Moreover, we have made more precise the relation between tip position and impact parameter that now takes into account the angle with the y-axis of initially radial trajectories, leading to almost invisible changes of the lines in Fig. 4. The details of the parameters (offset and deflection angle) used for the plots of Eq. (7) in Fig. 4 are now included in the text after Eqs. (7,8). The values are obtained by fitting the lines to the structures appearing in the figure.
  • The referee suggests more discussion of the effect of the tip shape. Following this advice, we have extended the corresponding parts in the text in the paragraph after Eq. (9) and in the Conclusions. There are indications that, in experiments, the tip potential is close to a Lorentzian shape, and we have added the new references [29,30] for this information. Such a shape is consistent with the conclusions of our analysis. The quantum simulations in Appendix A are made using a Gaussian tip shape in order to show that the effect is not limited to the particular situation of Lorentzian tip potentials and to confirm the importance of the mechanism by comparing the different forms of the features to the predictions of Sec. 4. Lines representing the result of Eq. (9) have been introduced in Fig. 11 in order to illustrate such a conclusion.
  • The referee says that he/she is used to see appendices as a way to provide technical details about a particular numerical or analytical computation or experimental setup. Admittedly, this is not the role of the appendices present in the manuscript. Appendices A and B stand for another perspective of what an appendix is supposed to be, and present alternative developments that are crucial in sustaining the picture presented in the main text, but would cut the flow of reading if they were placed elsewhere. They are sufficiently important, novel, and useful as to be part of the publication. While complementary to the main message presented in the text, they are of lesser relevance than the material therein discussed.

---

## Round 2 · Referee Report · Anonymous (Referee 3) · 2021-2-9

Strengths

1 - The paper clearly pinpoints a possible classical origin of some parabola-like contrast observed in recent experimental data.

2 - The analytical approach, combined with semiclassical simulations considering the exact experimental geometry, provides a convincing and simple framework to understand the physics discussed in the paper.

3 - The authors consider variations of different important parameters, tip shape, disorder realizations, on the observed phenomenon, a crucial point for comparison with experimental data.

Weaknesses

1 - The quantum simulations are not sufficiently exploited.

2 - The effect of tip shape (and the real tip shape in the experiment), which strongly influences the results, would deserve more discussion.

3 - Correspondance between semiclassical simulation results and Eq 7 (Fig. 4) seem to be somewhat "overoptimistic"

Report

I think that this represents a very nice and useful work, clearly pointing towards a classical phenomenon as the origin of parabolic-like features in recent experimental data from the group of T. Ihn. The paper is well written, and the discussion reads well, but there are still some significant weaknesses that can clearly be improved before publication.

Requested changes

1 - Compare the outcome of quantum simulations and (semi-)classical simulations : does the quantum framework also yield parabola-like features ?

2 - Gaussian/Lorentzian tip shape can be clearly distinguished from the shape of the resulting "parabola". The authors should elaborate more on this aspect, i.e. commenting on whether this could be helpful to identify the shape of the tip in experiments. In addition, very recent experimental work allowed to precisely measure the evolution of the SGM tip perturbation with the applied voltage in the "strongly invasive" regime in similar heterostructures (Appl. Phys. Lett. 117, 193101 (2020)), so the authors could rely on such data to provide a more realistic comparison with experimental data.

3 - Some of the "fits" to the data in Fig. 4 should be reduced to the ranges where there is indeed a good correspondance (e.g. some of the vertial portions of the dotted and continuous lines do not correspond to anything in the simulation data). It would also be helpful to know which parameters are changing between the different represented curves (I assume that E and \sigma are kept fixed ?)

4 - the discussion on the ideal cavity (without disorder) and of the different realization of disorder could be integrated in the paper, as these are important aspects.

  • validity: high
  • significance: good
  • originality: high
  • clarity: top
  • formatting: excellent
  • grammar: excellent

Author:  Dietmar Weinmann  on 2021-02-16  [id 1243]

(in reply to Report 3 on 2021-02-09)

We thank the referee for his/her positive overall judgment and helpful remarks. We have taken into account the suggestions as explained below.

  • The referee recommends a more thorough comparison between the quantum and classical simulations. We have presented such a direct comparison between the structures of the partial local density of states (Fig. 2) with the classical trajectory density (Fig. 3). At the level of the conductance, a discussion of the relation between quantum and classical simulations has been added in Sec. 3.1. of the revised version, where we also mention a detailed study comparing quantum and classical conductance as a function of tip strength in another cavity geometry (see Ref. [28] of the revised paper). The quantum calculations also exhibit arc-like structures. This important conclusion is now explicitly made by the dashed lines added in Fig. 11, thus complying with the first of the changes requested by the referee.
  • As expected from our model calculation of Sec. 4, the form of the arcs depends on the shape of the tip potential. The results shown in our Appendix A are calculated for Gaussian tip potentials in order to demonstrate the effect of the tip-potential shape along with the qualitative agreement. However, quantum calculations with different tip potentials in very similar systems are also presented in Ref. [18] of the revised version along with the experiment, and parabolic arcs appear there when Lorentzian tip potentials are used. This correspondence is mentioned in Sec. 3.2 of the revised version. Moreover, we now discuss this issue within a new paragraph at the end of Appendix A.
  • Following the suggestion of the referee, we have extended the discussion of the effect of the shape of the tip potential in the paragraph after Eq. (9) and in the Conclusions. We have included two new references [29,30] to papers with experimental methods to determine the tip potential and the size of the depletion disk.
  • We thank the referee for suggesting to remove the parts of the lines of the plots of Eq. (7) in Fig. 4 that are not expected to correspond to features in the conductance data. We have also specified the parameters that were used to draw the lines.
  • Concerning the appendices, though the information presented there is certainly important, we feel that putting them in the main part of the paper would cut the flow of the arguments containing the main message and thereby reduce the readability of the paper. They are nevertheless present and cross-linked to the main text whenever the additional information contained in the appendices is relevant.

---

## Round 3 · Referee Report · Anonymous (Referee 2) · 2021-2-18

Report
Most of the point I was raising in my first report have been addressed. In some circumstances, and especially for what concerns quantitative comparisons, this is done mainly by pointing to some reference to previous works but without adding new information on the actual problem under study, which is a bit frustrating. However the general point of view taken by the authors, and in particular their main result about the equi-conductance curves have been made significantly clearer. This is a nice paper including useful new information. It can be published in its present form.

---

## Round 3 · Author Response

Dear Editors,
We have revised and further improved our manuscript, taking into account all of the comments made by the referees. We herewith resubmit our manuscript for publication in SciPost Physics Core.
With best regards,
Dietmar Weinmann
We have revised and further improved our manuscript, taking into account all of the comments made by the referees. We herewith resubmit our manuscript for publication in SciPost Physics Core.
With best regards,
Dietmar Weinmann

---

## Round 3 · List of Changes

Text:
Introduction:
We gave a definition of what is understood by branches, and provided a footnote with a discussion about the difficulties in quantifying them. We included the new reference [7].
Sec. 3.1:
A discussion of the comparison between quantum and classical simulations has been added.
Sec. 3.2:
A footnote on the relation between experimental tip voltage and maximum potential height is added, towards helping to make a quantitative comparison with the experiment.
Sec. 4.:
We have changed the notation of the impact parameter \Delta x -> b, and added a discussion to take into account the precise relation between the impact parameter of radial non-vertical trajectories and the tip position. As a consequence, almost imperceptible changes of the lines in Fig. 4 occur. Accordingly, we have specified the precise parameters used in plotting those lines.
The discussion of effects of the shape of the tip potential has been extended, including the new references [29,30] pointing to experimental and theoretical studies of the tip-induced potential profile.
A discussion of the origin of the effect has been added at the end of the section, with the statement that the precise geometry of the cavity is not essential for the shape of the constant deflection-angle features.
Conclusions:
The discussion of the effect of the tip potential shape has been extended.
Appendix A:
A discussion of the presence of arc-like features and their form for different tip potential shapes is added at the end.
Figures:
Fig. 4: The lines representing Eq. (7) are slightly modified after taking into account the precise dependence of the impact parameter on the tip position. Units have been added to the vertical axis.
Fig. 5: Units have been added to vertical axis.
Figs. 7, 8: Consistently with the change in the text we have relabeled the impact parameter \Delta x -> b.
Fig. 11: We added dashed lines representing the prediction of Eq. (9).
Introduction:
We gave a definition of what is understood by branches, and provided a footnote with a discussion about the difficulties in quantifying them. We included the new reference [7].
Sec. 3.1:
A discussion of the comparison between quantum and classical simulations has been added.
Sec. 3.2:
A footnote on the relation between experimental tip voltage and maximum potential height is added, towards helping to make a quantitative comparison with the experiment.
Sec. 4.:
We have changed the notation of the impact parameter \Delta x -> b, and added a discussion to take into account the precise relation between the impact parameter of radial non-vertical trajectories and the tip position. As a consequence, almost imperceptible changes of the lines in Fig. 4 occur. Accordingly, we have specified the precise parameters used in plotting those lines.
The discussion of effects of the shape of the tip potential has been extended, including the new references [29,30] pointing to experimental and theoretical studies of the tip-induced potential profile.
A discussion of the origin of the effect has been added at the end of the section, with the statement that the precise geometry of the cavity is not essential for the shape of the constant deflection-angle features.
Conclusions:
The discussion of the effect of the tip potential shape has been extended.
Appendix A:
A discussion of the presence of arc-like features and their form for different tip potential shapes is added at the end.
Figures:
Fig. 4: The lines representing Eq. (7) are slightly modified after taking into account the precise dependence of the impact parameter on the tip position. Units have been added to the vertical axis.
Fig. 5: Units have been added to vertical axis.
Figs. 7, 8: Consistently with the change in the text we have relabeled the impact parameter \Delta x -> b.
Fig. 11: We added dashed lines representing the prediction of Eq. (9).

---

## Editorial Decision

resubmitted